# β-Catenin: A Key Molecule in Osteoblast Differentiation

**DOI:** 10.3390/biom15071043

**Published:** 2025-07-18

**Authors:** Edyta Wróbel, Piotr Wojdasiewicz, Agnieszka Mikulska, Dariusz Szukiewicz

**Affiliations:** Department of Biophysics, Physiology and Pathophysiology, Faculty of Health Sciences, Medical University of Warsaw, Chałubińskiego 5, 02-004 Warsaw, Poland; piotr.wojdasiewicz@wum.edu.pl (P.W.); agnieszka.mikulska@wum.edu.pl (A.M.); dariusz.szukiewicz@wum.edu.pl (D.S.)

**Keywords:** β-catenin, Wnt signaling, osteoblasts, differentiation of osteoblasts, bone

## Abstract

β-catenin is a key regulator of osteoblast differentiation, proliferation, and bone homeostasis. Through its interaction with transcription factors such as TCF/LEF, Runx2, and Osx, it coordinates gene expression essential for osteogenesis. The aim of this review is to demonstrate how β-catenin signaling is modulated by various physiological and pathological factors, including mechanical loading, oxidative stress, HIV-1 gp120, fluoride, implant topography, and microRNAs. These factors influence Wnt/β-catenin signaling through different mechanisms, often exerting opposing effects on osteoblast function. By integrating these modulators, we provide a comprehensive view of the dynamic regulation of β-catenin in bone biology. Understanding this complexity may provide insight into novel therapeutic strategies targeting β-catenin in bone regeneration, metabolic bone diseases, and pathologies such as HIV-associated bone loss or osteosarcoma.

## 1. Introduction

The Wnt/β-catenin signaling cascade, an evolutionarily conserved molecular network, is critically involved in a wide range of biological processes, from organogenesis and adult tissue equilibrium to the pathogenesis of numerous diseases [1,2]. Signal transduction is initiated when Wnt ligands bind to appropriate receptor complexes, activating intracellular pathways that are categorized based on the functional engagement of β-catenin. In the canonical signaling pathway, β-catenin acts as a key mediator, translocating to the nucleus to activate transcription of Wnt-responsive genes that regulate cell fate in various tissues. On the other hand, non-canonical Wnt signaling acts independently of β-catenin and involves mechanisms such as the planar cell polarization (Wnt/PCP) pathway and the protein kinase C (PKC)-dependent Wnt pathway. Currently, 19 Wnt proteins have been identified, each exerting effects through either canonical or non-canonical pathways. Notably, β-catenin serves functions beyond its role in Wnt-mediated signaling [3,4,5]. Initially characterized in the late 1980s, β-catenin was independently discovered for its dual roles in cell adhesion and signaling. The laboratory of Rolf Kemler isolated β-catenin along with α-catenin and γ-catenin/plakoglobin as E-cadherin-associated proteins dependent on calcium ions. These molecules were collectively termed “catenins” (from the Latin “catena,” meaning “chain”) due to their capacity to link E-cadherin to the actin cytoskeleton [6,7]. The transcriptional role of β-catenin was later elucidated through the study of its *Drosophila* homolog, Armadillo. Identified in genetic screens for mutations affecting embryonic segmentation by Wieschaus and colleagues, Armadillo’s function was shown to be under the regulatory influence of the Wingless gene, characterized in 1990 [8,9]. This pivotal discovery enabled the detailed characterization of the Wnt/β-catenin (or Wingless/Armadillo) pathway, delineating the signaling axis from the Wnt ligand, through Dishevelled, to the regulation of Armadillo protein stability via Shaggy/Zeste-white-3, the *Drosophila* homolog of glycogen synthase kinase 3 (GSK3) [10]. β-Catenin’s transcriptional activity in the nucleus is mediated by its interaction with members of the T cell factor/lymphoid enhancer factor (TCF/LEF) family of transcription factors, forming a complex that drives the expression of Wnt target genes [11]. Insights from mouse models with β-catenin gain-of-function and loss-of-function mutations have underscored its central role in regulating a spectrum of developmental processes across both embryonic and adult tissues [12]. Given the multifaceted nature of β-catenin function, the ability of this evolutionarily conserved signaling pathway to regulate a wide range of biological pathways during development and homeostasis depends on the dynamic modulation of its transcriptional activity. This regulation is mediated by a broad spectrum of β-catenin binding partners that influence its functional specificity and enable cross-talk with alternative signaling pathways and transcriptional regulators. These molecular interactions fine-tune β-catenin activity in a context-dependent manner, providing precise control of gene expression. A comprehensive analysis of these regulatory mechanisms is provided in the review by Valenta, Hausmann, and Basler [13,14].

In the context of skeletal biology, osteoblasts play a pivotal role in the development and preservation of bone structure. These cells mediate matrix deposition and modulate osteoclast activity. Canonical Wnt signaling is initiated upon the formation of a complex between Wnt ligands and Frizzled (FZD) receptors [15]. β-Catenin activity is indispensable for the terminal differentiation of osteoblasts and the subsequent promotion of bone formation. Interestingly, while β-catenin is not required for the initial transition of osteoprogenitor cells to early osteoblast precursors, its absence impedes the expression of the transcription factor Osterix (Osx), redirecting cell fate toward a chondrocytic lineage [16]. The low-density lipoprotein receptor-related protein 5 (LRP5) modulates bone mass through β-catenin-dependent Wnt signaling and has been shown in vitro to stimulate alkaline phosphatase (ALP), a marker of early osteoblast differentiation. Despite these advances, the specific Wnt ligand responsible for activating this pathway in osteoblastogenesis remains unidentified [16,17].

This review aims to elucidate the central role of β-catenin as a key regulatory molecule in the canonical Wnt signaling pathway, with particular emphasis on its function in controlling osteoblast differentiation and bone formation. Although the core components of this pathway have been extensively characterized, in our review, we characterize how β-catenin signaling in osteoblasts is modulated by a wide range of physiological and pathological stimuli, including mechanical loading, oxidative stress, viral agents such as HIV-1 gp120, fluoride exposure, implant surface topography, and microRNAs. These modulators represent distinct and, in many cases, opposing influences on Wnt/β-catenin signaling activity. By examining them in parallel, we highlight the multifactorial nature of osteogenic regulation and the dynamic adaptability of β-catenin signaling in response to environmental and molecular cues. This integrated approach offers new insights into the interplay of external stimuli and intracellular signaling mechanisms that ultimately govern osteoblast fate and function.

### Review Methodology

The literature search was conducted using PubMed, Scopus, and Web of Science databases. The following keywords were applied in various combinations: “β-catenin”, “osteoblast differentiation”, “Wnt signaling”, and “bone”. Relevant peer-reviewed articles were selected based on their focus on osteoblast biology, with particular emphasis on in vitro and in vivo studies addressing β-catenin function and regulatory mechanisms. Additional consideration was given to recent findings on the therapeutic modulation of Wnt/β-catenin signaling. Priority was given to original research articles and comprehensive reviews published within the last two decades.

## 2. Functional Implications of β-Catenin’s Molecular Structure

### 2.1. Molecular Architecture of β-Catenin

The crystal structure of β-catenin was described by Xing et al. in 2008 [18]. The β-catenin molecule consists of 781 amino acid residues in humans and includes a central region (residues 141–664), an N-terminal domain (NTD), an armadillo or ARM domain (comprising 12 amino acid repeats), a distal C-terminal domain (CTD) adjacent to the ARM domain, the terminal C-region, a C-helix, and unstructured sequences distal to the C-helix, each of which has specific functions. In the cytoplasm, β-catenin is targeted for degradation via NTD phosphorylation sites that enable interaction with components of the destruction complex (Axin, APC, and GSK3β) [6,19], whereas ligand interaction occurs mainly via the inner surface of the Armadillo (ARM) repeat domain. This domain mediates β-catenin binding to cadherins in the plasma membrane and transcription factors (e.g., TCF/LEF) in the nucleus [20]. The C-terminal domain (CTD) acts as a transactivation domain, interacting with transcriptional coactivators such as CBP/p300 ((CREB-binding)/E1A-associated protein p300) and recruiting chromatin remodeling complexes during transcriptional activation of Wnt target genes [21]. A key structural element in this region, the C-helix, is essential for canonical Wnt signaling, probably due to its role in attracting different coactivator proteins. The functional versatility of β-catenin is therefore intrinsically linked to its structural organization. While the specific binding motifs or amino acid residues involved in the interaction may vary depending on the signaling context, our current knowledge of how intrinsically disordered regions contribute to β-catenin activity remains incomplete. Therefore, further studies using advanced methodologies and innovative approaches are needed to elucidate the conformational dynamics of the β-catenin molecule [22,23]. As summarized in Table 1, the modular structure of β-catenin enables interactions with multiple binding partners at various subcellular locations [13,24,25,26,27].

Although β-catenin has been extensively studied in osteoblast biology, its domain-specific structural interactions remain less characterized in the context of bone physiology. β-catenin functions as a modular scaffold protein that facilitates distinct, context-dependent molecular interactions [28].

Our figure highlights how the modular structure of β-catenin facilitates distinct molecular interactions that are context-dependent (Figure 1). The N-terminal phosphorylation domain controls its cytoplasmic degradation, the armadillo repeat domain mediates cadherin binding and transcription complex assembly, and the C-terminal domain engages chromatin remodeling factors [29,30,31].

Understanding these domain interactions is crucial to explaining how β-catenin switches between adhesive and transcriptional roles during osteoblast differentiation and how its dysregulation contributes to bone pathologies such as osteoporosis and osteosarcoma [32,33].

Given the central roles that these complexes play in both physiological and pathological contexts, detailed characterization of the molecular architecture of β-catenin/cadherin and β-catenin/TCF (T-cell factor) complexes is of crucial importance. Remarkably, β-catenin in complex with cadherins has been implicated in tumor suppressive functions, whereas its association with TCF proteins has been implicated in promoting oncogenic transcriptional programs [22].

β-Catenin represents the prototypical member of the ARM superfamily. Its central domain is composed of a series of ARM repeats, each approximately 42 amino acids in length, forming a tri-helical motif with a characteristic triangular conformation. Collectively, these ARM repeats fold into a right-handed superhelical structure, distinguished by an extended, positively charged groove along its inner surface [31].

Structural and biochemical investigations, including high-resolution crystallography, have demonstrated that a diverse array of β-catenin-interacting proteins bind within this groove, frequently occupying overlapping or adjacent sites, thereby indicating a conserved mechanism of molecular recognition within the ARM domain [34]. This means that the molecules cannot bind to β-catenin simultaneously [35].

This mutual exclusivity is certainly important for key molecules interacting with β-catenin, such as E-cadherin (a major partner in adherens junctions) [36], APC (a major partner in the destruction complex), and TCF/LEF (a major partner in the nucleus) [37].

The ability of β-catenin to associate with multiple partners at both the plasma membrane and the nucleus is regulated by the structural organization of its central ARM domain, as well as by the kinetics and affinity of competing protein interactions [26].

β-catenin-binding partners are located within the core region comprising ARM repeats 3 through 9 (R3–R9), where conserved salt bridges are formed with two critical lysine residues, Lys312 and Lys435 [13,38]. Crystallographic studies have shown that many β-catenin interactors, including cadherins and the transcription factors TCF/LEF, bind to overlapping regions in the ARM repeats [39].

The spatial organization of β-catenin’s binding surfaces plays a crucial role in ensuring the selective engagement of distinct molecular partners within the cellular environment. Competitive binding among these interacting proteins is particularly significant for the modulation of Wnt/β-catenin signaling activity as it can influence downstream transcriptional outcomes [24]. Moreover, the conformational plasticity of the β-catenin molecule may contribute to the regulation of its interaction profile. Structural studies suggest that both the NTD and, more prominently, the CTD can fold back onto the central ARM domain, a rearrangement that may have an impact on the accessibility of key interaction sites, such as those involved in TCF/LEF transcription factor binding [40].

Under physiological conditions, partner prioritization is driven primarily by binding affinity, local concentration, and compartmentalization. For example, the cytoplasmic tail of cadherin binds β-catenin with high affinity (Kd ~10–50 nM), effectively sequestering it at the plasma membrane in the absence of Wnt signaling [41]. In contrast, nuclear partners such as TCF/LEF family members have lower affinity interactions, but these are stabilized under conditions in which β-catenin accumulates in the cytoplasm due to Wnt pathway activation or cadherin uncoupling [12].

Pathological conditions such as cancer often disrupt this regulatory balance. Mutations in β-catenin (CTNNB1) or key components of the destruction complex, such as APC or Axin, result in constitutive stabilization of β-catenin, allowing its accumulation in the cytoplasm and subsequent translocation into the nucleus, where it drives aberrant transcriptional programs [42,43]. In addition, during tumor progression, the loss or proteolytic cleavage of cadherins diminishes the ability of the plasma membrane to retain β-catenin, thereby increasing the cytoplasmic pool available for nuclear signaling. These alterations shift the functional balance of β-catenin from maintaining cell–cell adhesion toward promoting oncogenic gene expression [36,44].

### 2.2. β-Catenin and Its Interactions with Adhesion Molecule

Biochemical evidence supports a model in which transcriptional activation by β-catenin is mediated by its monomeric form, characterized by a backward-folded conformation. In contrast, the dimeric conformation of β-catenin, which engages in cadherin binding, is functionally linked to α-catenin at adherens junctions, where it contributes to intercellular adhesion [25]. Notably, the C-terminal domain of β-catenin contains a critical structural element Helix-C positioned at its extreme N-terminal portion. This helix has been demonstrated to be indispensable for β-catenin’s transcriptional signaling function; however, it appears to be dispensable for its role in mediating cell–cell adhesion [6,13]. In fact, many β-catenin transcriptional coactivators require an intact Helix-C for proper binding [45].

Studies in the late 1990s showed that mutations in the Armadillo fragment disrupted adhesive function but had no effect on Wnt signaling [46,47]. Other studies have shown that Armadillo has the ability to completely suppress signaling output without affecting its adhesive role [13]. In the majority of metazoan species, including established model organisms such as Drosophila melanogaster, Xenopus laevis, and Mus musculus, a single β-catenin protein fulfills dual roles. These distinct functional modalities are thought to be executed by separate pools of β-catenin, whose distribution and activity are tightly regulated through mechanisms such as post-translational modifications that govern their spatial localization, stability, and retention within the cell [48].

Under basal conditions, the majority of β-catenin is localized at the cytoplasmic face of the membrane, where it integrates into cadherin-dependent cell adhesion complexes. This cadherin–catenin complex constitutes the molecular core of adherens junctions, which are essential for maintaining intercellular cohesion and enabling coordinated cell behavior across both epithelial and mesenchymal contexts. By physically linking adjacent cells to the underlying actin–myosin cytoskeletal network, the cadherin-based adhesion system facilitates mechanical coupling between cells. This linkage is fundamental for a range of biological processes, including morphogenetic movements during development and the restoration of tissue integrity following injury [49]. Cadherins are transmembrane glycoproteins that mediate intercellular adhesion in the presence of *extracellular calcium* [50].

*Cadherin-mediated cell–cell adhesion* depends primarily on homotypic interactions between identical cadherin molecules, suggesting a Ca^2+^-dependent adhesive function via their extracellular regions that also interact with β-catenin through their cytoplasmic tails [43,51,52,53]. Classical cadherins derive their nomenclature from the tissues in which they were first identified or are predominantly expressed. For instance, E-cadherin (epithelial cadherin) is characteristic of epithelial tissues [54], N-cadherin (neural cadherin) is primarily found in the nervous system, and M-cadherin (muscle cadherin) is associated with muscle tissue [55,56]. β-Catenin engages directly with the cytoplasmic tails of each of these cadherin subtypes, forming a key component of the cadherin–catenin complex.

During biosynthesis, E-cadherin interacts with β-catenin early in the secretory pathway, with complex formation occurring within the endoplasmic reticulum. Following this interaction, the cadherin–β-catenin complex is trafficked to the plasma membrane. Importantly, interference with this interaction such as through genetic or biochemical disruption leads to proteasomal degradation of the cadherin molecule, underscoring the stabilizing role of β-catenin in cadherin turnover and cell adhesion maintenance [57].

### 2.3. β-Catenin’s Involvement in Cytoplasmic Interactions

The interaction between E-cadherin molecules contributes to the stabilization of β-catenin by obstructing the association of β-catenin with components of its degradation machinery, specifically the adenomatous polyposis coli protein (APC) and Axin (cytosolic scaffold protein) [26]. The major adhesive interactions are those between the extracellular (EC) domains N-terminal 1 (EC1). At the same time, the cytoplasmic domains interact with the cytoskeleton by binding catenins, thus providing a link between adhesion events and intracellular dynamics. Ultimately, through its association with the cytoskeleton, E-cadherin activates signals that regulate cell shape, polarity, proliferation, and differentiation [57,58].

In the absence of extracellular Wnt ligands, the canonical Wnt pathway is inactive. Cytoplasmic β-catenin is recognized by the destruction complex and thus targeted for degradation (Figure 2). The destruction complex includes the tumor suppressor proteins Axin and adenomatous polyposis coli (APC), as well as casein kinase 1α (CK1α) and glycogen synthase kinase 3β (GSK3β). These kinases phosphorylate β-catenin, thereby inducing proteasomal degradation. In the nucleus, TCF and LEF proteins interact with transcriptional corepressors of the Groucho/TLE family to block the expression of Wnt target genes. Activation of the pathway is usually mediated by Wnt ligands that bind to transmembrane receptors such as the G protein-coupled receptors FZD and lipoprotein receptor-related protein 5/6 (LRP5/6), thereby triggering the recruitment of Dishevelled (Dvl) to the plasma membrane [2,38,59].

This leads to the binding and inhibition of the destruction complex and thus the accumulation and nuclear translocation of β-catenin. In the nucleus, β-catenin binds to transcription factors of the TCF/LEF family, resulting in the transcription of Wnt target genes [38].

### 2.4. β-Catenin’s Nuclear Activity and Its Implications

In the cytoplasmic compartment, β-catenin exhibits a short half-life due to its rapid recognition and targeting by the destruction complex, which facilitates its proteolytic degradation. This regulatory mechanism applies to both newly synthesized β-catenin as well as to molecules released from adherens junctions. Cytoplasmic β-catenin associates with Axin and APC, forming a multiprotein complex in which interactions may occur either directly or through intermediary factors. This complex functions as a molecular scaffold that enables the recruitment of kinases responsible for sequential phosphorylation of β-catenin [60].

The initial phosphorylation event is mediated by CK1α, which targets Ser45. This primes β-catenin for subsequent phosphorylation by GSK3, predominantly GSK3β, at Thr41, Ser37, and Ser33. Following these phosphorylation events, the APC-associated β-catenin is released from the destruction complex and engaged by the ubiquitination machinery. Specifically, β-catenin phosphorylated at Ser33 and Ser37 is recognized by the E3 ubiquitin ligase adaptor β-TrCP, which recruits it to the Skp1/Cul1/F-box/β-TrCP (SCFβ-TrCP) E3 ligase complex. The polyubiquitinated β-catenin is subsequently directed to the proteasome for degradation [61].

The scaffolding proteins Axin and APC are indispensable for GSK3-dependent phosphorylation of β-catenin, as GSK3 is catalytically active in this context only when β-catenin is presented within the Axin–APC scaffold. Notably, Axin, often considered the rate-limiting component of the destruction complex, significantly enhances GSK3 activity toward β-catenin. APC, in turn, plays a crucial role in both the structural assembly of the complex and in maintaining the phosphorylated state of β-catenin, thereby promoting its efficient degradation [62].

Upon leaving the destruction complex, N-terminally phosphorylated β-catenin, if not bound to APC, is promptly dephosphorylated by PP2A (Protein Phosphatase 2A). A mutation in APC that disrupts this protective function leads to the exposure of the N-terminal phosphorylated serine/threonine residues of β-catenin to PP2A [63].

β-Catenin plays dual cellular roles: it acts as a structural component of adherens junctions at the plasma membrane and as a transcriptional co-activator in the nucleus. The ability of β-catenin to switch between these functions is tightly regulated by its post-translational modifications and by spatial compartmentalization within the cell. At the plasma membrane, β-catenin binds directly to the cytoplasmic tail of cadherins, stabilizing cell–cell adhesion and linking the cadherin complex to the actin cytoskeleton. This membrane-associated β-catenin pool is relatively stable and protected from the degradative mechanisms acting on the cytoplasmic form of the protein. Importantly, this adhesive role is largely independent of Wnt/β-catenin signaling [64,65].

Additional post-translational modifications further regulate this functional switch. Tyrosine phosphorylation, particularly at Tyr654, reduces β-catenin’s affinity for cadherins, promoting its dissociation from adherens junctions and enhancing its availability for nuclear translocation and transcriptional activity. In the nucleus, acetylation by CBP/p300 enhances β-catenin’s transcriptional co-activator function by stabilizing its interaction with TCF/LEF transcription factors and facilitating chromatin remodeling [66]. Ubiquitination, primarily targeting phosphorylated β-catenin in the cytoplasm, remains a key regulatory mechanism for controlling its degradation and preventing aberrant nuclear accumulation [19].

The functional versatility of β-catenin relies on its ability to adopt distinct conformations and engage with different binding partners depending on its subcellular localization and post-translational modification status. Cara Gottardi and Barry Gumbiner provided seminal insights demonstrating that Wnt pathway activation induces conformational rearrangements in β-catenin that favor its assembly into nuclear transcription complexes, promoting gene expression [25]. In contrast, β-catenin associated with α-catenin at the plasma membrane adopts a conformation that stabilizes cadherin mediated cell–cell adhesion, preventing its nuclear translocation and transcriptional activity [25,27].

Further insights were provided by Felix Brembeck and colleagues, who showed that post-translational phosphorylation of β-catenin at specific N-terminal residues modulates its functional compartmentalization. Phosphorylation disrupts β-catenin’s association with the cadherin–catenin adhesion complex, thereby reducing its membrane localization. Concurrently, these phosphorylation events enhance β-catenin’s ability to bind nuclear partners, such as TCF/LEF transcription factors, facilitating its transcriptional co-activator role [44].

Collectively, these findings highlight a sophisticated regulatory system wherein context-dependent β-catenin modifications, interacting protein partners, and subcellular localization collectively determine whether β-catenin functions predominantly in maintaining cell–cell adhesion or driving Wnt-dependent gene transcription (Figure 3). This dynamic regulation is essential for osteoblast proliferation and differentiation, as well as for maintaining skeletal homeostasis. Dysregulation of these processes contributes to bone pathologies, including osteoporosis and osteosarcoma, where altered β-catenin signaling disrupts the balance between osteogenesis and cellular adhesion integrity [42,67].

## 3. Wnt/β-Catenin Pathway in Bone Formation

### 3.1. Major Molecules Involved in Canonical Signaling in Bone

Wnt proteins initiate a cascade of intracellular signaling events that regulate diverse cellular functions, including cell fate specification, proliferation, migration, polarity establishment, and gene transcription. These ligands activate three major signaling pathways: the canonical Wnt/β-catenin pathway, the Wnt/Ca^2+^ pathway, and the planar cell polarity (PCP) pathway. Among them, the Wnt/β-catenin pathway, commonly referred to as the canonical pathway, plays a central role in promoting cell proliferation, survival, and fate determination by stabilizing β-catenin and modulating gene expression via TCF/LEF transcription factors. Activation of the canonical Wnt pathway begins with the binding of Wnt ligands to FZD receptors and their co-receptors, LRP5/6 [68].

In the absence of Wnt ligands, cytoplasmic β-catenin is phosphorylated by a destruction complex, marking it for ubiquitin-mediated proteasomal degradation and thereby preventing the activation of Wnt target genes [15,69]. The pathway is tightly regulated by a variety of extracellular antagonists. These include sclerostin (SOST), Dickkopf (DKK) family members (DKK1, DKK2, and DKK3), and secreted frizzled-related protein 1 (sFRP1). These inhibitors either sequester Wnt ligands (e.g., sFRP1) or interfere with Wnt–receptor interactions (e.g., sclerostin and DKK1), thereby preventing β-catenin stabilization and nuclear translocation [23]. Aberrant activation of Wnt/β-catenin signaling is implicated in the pathogenesis of numerous diseases, including skeletal disorders such as osteoporosis [70].

In contrast, the non-canonical Wnt signaling pathways operate independently of β-catenin stabilization and primarily regulate processes such as PCP, cell migration, and intracellular calcium dynamics. The Wnt/PCP pathway activates small Rho family GTPases (e.g., RhoA, Rac1) and c-Jun N-terminal kinase (JNK), orchestrating cytoskeletal organization and cellular orientation [7]. Meanwhile, the Wnt/Ca^2+^ pathway triggers intracellular calcium release via activation of phospholipase C (PLC), leading to the stimulation of calcium-dependent effectors such as calmodulin-dependent kinases, calcineurin, and PKC [68].

Notably, Wnt5a and Wnt11 are prominent ligands that preferentially activate non-canonical signaling. Moreover, crosstalk between canonical and non-canonical pathways is evident, with non-canonical signals often antagonizing β-catenin-mediated transcription, thereby fine-tuning cell fate decisions during skeletal development [71].

The functional divergence between these pathways underscores the complexity of Wnt signaling in regulating osteogenesis and highlights β-catenin as a pivotal node in both physiological bone formation and pathological conditions such as osteoporosis and osteosarcoma [72].

The link between Wnt signaling and bone homeostasis was established through genetic studies showing that inactivating mutations in the LRP5 gene lead to reduced bone mass and osteoporosis in humans. LRP5 is expressed at relatively low levels across various tissues with minimal temporal variation, yet it plays a critical regulatory role in skeletal physiology. Its function is closely tied to its ability to form a receptor complex with Frizzled and Wnt proteins, thereby facilitating the activation of the canonical pathway (Table 2). β-Catenin activity is indispensable for the terminal differentiation of osteoblasts and subsequent bone formation [73].

Interestingly, while β-catenin is not required for the initial differentiation of osteoprogenitor cells into pre-osteoblasts, its absence blocks the expression Osx redirecting the cell fate toward a chondrogenic lineage. The regulatory influence of LRP5 on bone mass is mediated through β-catenin signaling and, in vitro, is associated with the induction of ALP, an early marker of osteoblast differentiation [74]. Several Wnt ligands, including Wnt1, Wnt4, Wnt5α, Wnt7b, and Wnt9α/14, are expressed in osteoblast precursors. Conversely, Wnt3α and Wnt10b are localized primarily in the bone marrow; among these, only Wnt10b-deficient models exhibit postnatal bone loss. In vitro, Wnt10b expression in mesenchymal progenitor cells induces osteoblastogenic transcription factors such as core-binding factor α1 (Cbfa1/Runx2), distal-less homeobox 5 (Dlx5), and Osx, reinforcing the role of Wnt signaling in promoting osteoblast differentiation [75].

Although canonical Wnt ligands converge on β-catenin stabilization and nuclear translocation, their downstream transcriptional outputs are surprisingly context-dependent in osteoblasts. Several molecular mechanisms contribute to this specificity [3].

β-catenin does not act alone but forms transcriptional complexes with various cofactors, the composition of which varies depending on the cellular context. In osteoblasts, β-catenin preferentially interacts with TCF7, LEF1, and coactivators such as p300/CBP and BCL9 (B-cell lymphoma 9), modulating the expression of osteoblast-specific genes such as Runx2, Osterix, and ALP [1,2]. More, epigenetic factors regulate the chromatin accessibility of Wnt target genes. Histone acetyltransferases (e.g., CBP/p300) and histone methyltransferases (e.g., MLL1/2 complexes (Mixed-lineage leukemia 1 and 2)) recruited by β-catenin complexes promote transcriptional activation of osteogenic genes, whereas the presence of corepressors such as Groucho (in Drosophila)/TLE (Transducin-Like Enhancer) can repress alternative targets of β-catenin [3,4,76,77]. The availability of lineage-determining transcription factors (e.g., Runx2, Dlx5, ATF4) in osteoblasts further directs β-catenin activity toward osteogenesis rather than alternative Wnt-regulated processes such as cell proliferation or migration [78]. Finally, different Wnt ligands may preferentially engage different combinations of Frizzled receptors and LRP5/6 co-receptors, resulting in the formation of ligand-specific signalosomes and modulation of the amplitude and duration of β-catenin stabilization [79].

Additionally, Wnt10b negatively regulates adipogenesis by inhibiting transcription factors such as CCAAT/enhancer-binding protein α (C/EBPα) and peroxisome proliferator-activated receptor γ (PPARγ) [80,81]. This illustrates the reciprocal relationship between *osteoblast* and *adipocyte* lineage commitment, whereby enhanced osteogenic differentiation is typically associated with suppressed adipogenesis [82]. Indeed, in the absence of instructive cues from signaling pathways like Wnt, mesenchymal progenitors may default toward an adipocytic fate [83]. Adipogenesis is most likely the default pathway for cells that do not receive appropriate inductive *signals* to become osteoblasts, chondrocytes, myocytes, or other mesodermal cell types [84].

In fact, there is a trade-off between osteoblast and adipocyte differentiation, with an increase in osteoblast differentiation being associated with a decrease in adipocyte differentiation [15].

Regulation of the Wnt pathway also involves inhibition at the receptor level. The interaction between Wnt ligands and FDZ is competitively blocked by secreted Wnt inhibitors, including the sFRPs nd Wnt inhibitory factor 1 (WIF 1) [85,86]. Meanwhile, LRP5/6 activity is antagonized by sclerostin and DKK family proteins, providing an additional layer of control over pathway activation [87] (see Figure 4).

Activation of the canonical Wnt signaling cascade leads to upregulation of Dickkopf-2 (Dkk2), a downstream effector that contributes to the terminal differentiation of osteoblasts. Dkk2 facilitates this process by promoting cell cycle exit, thereby enabling maturation and matrix mineralization. Importantly, Dkk2 expression increases during the later stages of osteogenesis and is required for the formation of mineralized bone tissue [88].

LRP5 and LRP6 function as coreceptors for Wnt ligands and also mediate signaling initiated by other molecules such as sclerostin. Sclerostin is produced primarily by osteocytes and exerts a negative regulatory effect on osteoblast biology, inhibiting their proliferation and differentiation while promoting apoptotic processes. Loss-of-function mutations that reduce SOST (the gene encoding sclerostin) expression in humans lead to excessive bone formation, as observed in the high bone mass disorder known as sclerosteosis. The function of mature osteoblasts, particularly their capacity to synthesize extracellular matrix components, requires both LRP5 and the activity of activating transcription factor 4 (ATF4), a key regulator of osteoblastic gene expression [16]. Beyond its role in osteoblastogenesis, the Wnt/β-catenin pathway also modulates bone remodeling by inhibiting osteoclastogenesis. This is achieved, at least in part, through the upregulation of osteoprotegerin (OPG), a decoy receptor that prevents receptor activator of nuclear factor κB ligand (RANKL) from promoting osteoclast differentiation [89].

Experimental evidence from in vitro studies indicates that OPG transcription is directly regulated by β-catenin signaling. In osteoblasts, the β-catenin–TCF complex binds to the OPG promoter, and among TCF family members, TCF1 appears to play a central role in this transcriptional control [14,89].

**Table 2 biomolecules-15-01043-t002:** Core components and modulators of the canonical Wnt/β-catenin signaling pathway and their functions. Proteins were selected based on their essential role in Wnt ligand-receptor interactions, β-catenin stabilization/degradation, or transcriptional activation, as documented in primary research and review articles.

Proteins	Function in the Wnt/β-Catenin Signaling Pathway	Reference
APC	Forms the β-catenin destruction complex with CK1, AXIN, and GSK3β	[42] Clevers & Nusse, 2012
AXIN	Scaffold protein in the β-catenin destruction complex with CK1, APC, and GSK3β	[62] Stamos & Weis, 2013
β-catenin	Central mediator of the Wnt/β-catenin pathway; translocates to the nucleus to regulate TCF/LEF-dependent gene transcription	[45] MacDonald et al., 2009
CK1α	Participates in the phosphorylation of β-catenin within the destruction complex	[90] Amit et al., 2002
DKKs	Extracellular antagonists blocking Wnt-FZD-LRP5/6 complex formation	[91] Niehrs, 2006
DVL	Cytoplasmic protein transmitting Wnt signals from FZD-LRP5/6 to downstream effectors	[92] Gammons & Bienz, 2018
FZD	Cell surface receptor for Wnt ligands, initiating pathway activation	[93] Schulte, 2010
GSK3β	Kinase phosphorylating β-catenin (Thr41, Ser33, Ser37), marking it for degradation	[94] Wu & Pan, 2010
LRP5/6	Co-receptors forming a complex with FZD to bind Wnt ligands	[95] He et al., 2004
PP2A	Dephosphorylates β-catenin, regulating its stability	[42] Clevers & Nusse, 2012
sFRP	Extracellular inhibitors sequestering Wnt ligands, preventing receptor binding	[20] Cadigan & Waterman, 2012
TCF/LEF	Nuclear transcription factors that partner with β-catenin to activate gene expression	[20] Cadigan & Waterman, 2012
WIF1	Binds Wnt ligands, preventing interaction with FZD-LRP5/6 receptors	[96] Hsieh et al., 1999
Wnt	Secreted ligands activating the canonical Wnt/β-catenin signaling cascade	[59] Nusse & Clevers, 2017

### 3.2. Wnt/β-Catenin Ppthway as a Regulator of Osteoblast Differentiation

The skeletal system is primarily composed of bone tissue, which includes three principal cell types: osteoblasts, osteoclasts, and osteocytes [97]. In adults, bone mass homeostasis is tightly regulated by the dynamic equilibrium between bone matrix synthesis by osteoblasts and bone resorption by osteoclasts. Disruption of this balance leads to pathological conditions characterized by accelerated bone loss, such as osteoporosis [98,99,100].

Among the critical regulators of this balance are Wnt ligands, which orchestrate the differentiation and activity of both osteoblasts and osteoclasts within the bone marrow microenvironment (Figure 4). Canonical Wnt/β-catenin signaling is particularly important in directing the lineage commitment of mesenchymal progenitor cells during both embryonic development and adult tissue maintenance [101]. During embryogenesis, skeletal tissue develops through both endochondral and intramembranous ossification pathways. The Wnt/β-catenin pathway promotes osteogenic over chondrogenic or adipogenic differentiation by inhibiting the commitment of mesenchymal progenitors to chondrocyte and adipocyte lineages while simultaneously promoting osteoblastogenesis [102]. Wnt/β-catenin signaling initiates osteogenic differentiation in mesenchymal precursors and guides osteochondral progenitors toward the osteoblastic lineage. This pathway also contributes to the proliferation, maintenance, functional regulation, and survival of osteoblasts and osteocytes. Reduced Wnt signaling activity has been associated with increased production of pro-apoptotic factors in these cells, underscoring its role in cellular longevity [102,103].

In addition to its role in lineage specification, Wnt signaling modulates the transcription of genes necessary for bone matrix production and mineralization. Canonical Wnt signaling upregulates alkaline phosphatase expression in pluripotent mesenchymal cell lines, while Wnt3a specifically enhances the expression of type I collagen and osteopontin two critical components of the bone matrix [104].

Wnt1, Wnt2, and Wnt3a have also been shown to induce ALP expression and simultaneously suppress the expression of PPARγ, thereby inhibiting adipogenic differentiation in favor of osteoblast formation [105,106,107].

Osteoblasts also play a crucial role in the regulation of osteoclastogenesis. They express RANKL, which binds to its receptor RANK on osteoclast precursors, promoting their differentiation. In parallel, osteoblasts secrete OPG, a decoy receptor that binds RANKL, preventing its interaction with RANK and thereby inhibiting osteoclast development and activity [108,109]. Furthermore, osteoblasts secrete various ligands and factors that regulate hematopoietic stem cell (HSC) maintenance, including self-renewal and proliferation. Genetic studies have revealed the importance of Wnt/β-catenin signaling in skeletal development and maintenance. Loss-of-function mutations in the LRP5 gene are associated with diminished bone formation and decreased bone mineral density, whereas gain-of-function mutations result in increased bone mass in both humans and mouse models, highlighting the pathway’s role in promoting osteogenesis. Inactivation of β-catenin in early mesenchymal progenitors results in impaired osteoblast differentiation and skeletal defects. However, deletion of β-catenin in mature osteoblasts or during late-stage osteocyte differentiation does not impact bone formation but leads to significantly enhanced osteoclastogenesis [109,110,111].

Quanwei Bao’s research team showed results where constitutive activation of β-catenin (CA-β-catenin) in osteoblasts potentially has side effects on bone growth and bone remodeling process, although it can increase bone mass. The present study aimed to observe the effects of osteoblastic CA-β-catenin on bone quality and explore the possible mechanisms of these effects. It was found that CA-β-catenin mice showed lower mineralization levels and disorganized collagen in long bones, which was confirmed by von Kossa staining and Sirius red staining, respectively [112].

Thus, the Wnt/β-catenin pathway appears to be a positive regulator of osteoblast differentiation. The Wnt/β-catenin pathway is responsible for the development of osteoblast lineages from precursors, not chondroblasts. In mature osteoblasts, this pathway produces signals that inhibit osteoblast differentiation. β-Catenin functions downstream of the bone morphogenetic protein (BMP) signaling pathway, which plays a pivotal role in skeletal development and bone formation [113].

Canonical Wnt signaling has been shown to promote the expression of several BMP family members, including BMP2, BMP4, and BMP7. The interplay between Wnt/β-catenin and BMP signaling pathways contributes to the regulation of osteoblast differentiation and bone matrix production. Despite extensive evidence supporting the cooperative roles of these pathways, the precise molecular mechanisms underlying their interaction during osteogenesis remain incompletely understood. Nonetheless, experimental studies have demonstrated that simultaneous activation of β-catenin signaling and deletion of Smad4, an essential mediator of canonical BMP signaling, results in enhanced osteoblast proliferation. This finding suggests that β-catenin and Smad4 may exert antagonistic effects on the regulation of osteoblast proliferation and differentiation, highlighting a complex balance between proliferative and differentiative cues during bone tissue development [113,114].

Wnt10b is a key activator of the canonical Wnt/β-catenin signaling pathway and has been shown to promote osteoblast differentiation. Similarly, activation of this pathway through Wnt3a or pharmacological inhibition of GSK3β using lithium suppresses dexamethasone-induced osteogenesis in human mesenchymal stem cells (hMSCs) [115].

Extensive evidence from the literature indicates that canonical Wnt signaling facilitates osteoblastogenesis by upregulating the transcription of osteogenic markers such as Runx2, Dlx5, and Osx in pluripotent mesenchymal and osteoprogenitor cells in murine models [107]. The transcription factor Runx2 is indispensable for the initiation of osteoblast differentiation. However, its activity is negatively regulated by GSK3β in osteoblastic precursors. Inhibition of GSK3β, an event promoted by Wnt/β-catenin signaling, leads to Runx2 activation, thereby supporting osteoblast proliferation and differentiation [116,117].

Pharmacological suppression of GSK3β has been shown to enhance osteogenesis in vitro, further underscoring the central role of this kinase in controlling osteogenic commitment [118,119]. As highlighted in the review by Ahmadzadeh, several studies have also reported increased expression of extracellular Wnt inhibitors, including Dkk1, Dkk2, sFRP2, and sclerostin, during the later stages of osteoblast differentiation. These findings suggest that Wnt signaling not only promotes early osteoblastogenesis but may also play a crucial role in coordinating the terminal maturation and mineralization of the extracellular matrix by osteoblasts [107,120,121].

Moreover, genetic alterations affecting components of the Wnt signaling cascade further support its role in skeletal development. Mutations in the Axin gene and null mutations in sFRP1, as well as disruptions in other regulatory elements of the Wnt/β-catenin pathway, have been associated with enhanced osteoblast differentiation and increased bone mass, accompanied by greater matrix mineralization [122]. Numerous studies support the involvement of Wnt/β-catenin signaling in the process of osteoblast differentiation, although the mechanisms are still controversial. Liang et al. showed that the downregulation of docking protein 5 (Dok5), a recently discovered cytosolic tyrosine kinase-related signaling molecule Doks, inhibited osteoblastic differentiation as well as the expression of osteogenic biosignatures. Conversely, overexpression of Dok5 promotes proliferation and osteogenesis and activates canonical Wnt/β-catenin signaling. Thus, Dok5 may regulate proliferation and osteogenic differentiation via this pathway [123].

## 4. Modulators of Canonical Wnt/β-Catenin Pathway in Osteoblast Function and Bone Tumorigenesis: Role of Intracellular, miRNAs, Mechanical Loading, and Extracellular Signals

### 4.1. Intracellular Regulators of β-Catenin in Osteoblasts

During osteogenic differentiation of the pre-osteoblast-like mouse cell line MC3T3-E1, CDK14 (Cyclin-dependent kinase 14) was shown to play a key role in modulating Wnt/β-catenin signaling. CDK14 promotes the phosphorylation of low-density lipoprotein receptor protein 6 (LRP6) and glycogen synthase kinase-3β (GSK3β), two key components of the β-catenin regulatory cascade, thereby facilitating β-catenin stabilization and nuclear translocation [124].

Importantly, CDK14 expression was found to be markedly reduced in bone tissue samples derived from a mouse model of postmenopausal osteoporosis, suggesting that its downregulation may contribute to impaired β-catenin signaling in osteoporotic conditions. Moreover, pharmacological inhibition or genetic silencing of CDK14 in osteoblast precursors altered cell cycle progression, reduced proliferation rate, and significantly inhibited osteogenic differentiation, as evidenced by reduced ALP activity and mineralization potential [124].

These findings provide additional insight into the molecular networks controlling β-catenin-dependent osteoblast proliferation, lineage commitment, and the pathophysiology of osteoporosis [125]. Furthermore, lamin A, a structural component of nuclear lamin, has been implicated in osteoblast differentiation via interaction with β-catenin signaling. In a study by Tsukune et al. a mutant form of mouse lamin A, called lamin A dC50, encoding a truncated version analogous to progerin, was transfected into MC3T3-E1 cells [126].

This mutant form of lamin A led to reduced cytoplasmic β-catenin activity due to increased interaction with GSK3β, resulting in inhibition of β-catenin signaling. Consequently, osteoblast differentiation was inhibited at both early and late stages. The authors reported a significant reduction in ALP activity, as well as in the mRNA expression levels of essential osteogenic markers, including type I collagen (COL1), bone sialoprotein (BSP), and Runx2, all of which are key effectors of β-catenin-mediated osteoblast differentiation [127,128]. These results underscore the importance of nuclear envelope integrity in maintaining β-catenin signaling homeostasis during osteoblastogenesis [127,129].

### 4.2. Role of β-Catenin in Mesenchymal and Other Progenitor Cell Lineages

In murine bone marrow mesenchymal stem cells (BMSCs), exposure to hydrogen peroxide (H_2_O_2_) leads to a marked reduction in the expression levels of telomerase reverse transcriptase (TERT)-associated pathway components, including β-catenin and the phosphorylated/inactive form of GSK3β (p-GSK3β), relative to total GSK3β. This downregulation is accompanied by decreased expression of osteogenic markers and anti-apoptotic proteins, along with an upregulation of pro-apoptotic proteins, when compared to untreated controls. Mechanistic investigations have revealed that activation of the canonical Wnt signaling pathway in BMSCs enhances nuclear translocation of β-catenin, a key event required for the transcriptional regulation of osteogenic genes. These findings suggest that the GSK3β/β-catenin/TERT axis plays a critical role in modulating both osteogenic differentiation and apoptotic responses in BMSCs. Consequently, this signaling pathway may represent a promising therapeutic target for the treatment of osteoporosis [130].

Other studies indicate that reducing β-catenin levels in skeletal muscle satellite cells (SMSCs) leads to lower bone mass and deteriorated bone microstructure in vivo, as well as delay fracture healing; while, at the same time, in vitro culture experiments of SMSCs found that their osteogenic differentiation capacity was reduced [131,132].

Similar β-catenin-dependent mechanisms have been described in other progenitor cell populations. For instance, conditional deletion of β-catenin in mesenchymal stem cells (MSCs) impairs their osteogenic differentiation while favoring adipogenic lineage commitment, leading to reduced bone formation and increased marrow fat accumulation in vivo [97,133]. Furthermore, β-catenin deficiency in MSCs was shown to suppress the expression of key osteogenic transcription factors, including Runx2, Osterix, and ALP, and attenuate mineralization capacity in vitro [134,135].

Moreover, in a study involving bone marrow stromal cells (BMSCs), β-catenin knockdown inhibited their ability to form bone when transplanted into ectopic sites in immunodeficient mice, confirming its essential role in osteoblast lineage commitment and bone tissue formation [97,136]. β-catenin signaling has also been shown to regulate periosteal progenitor cells, another critical osteogenic cell population. Mice with β-catenin deletion in these cells exhibited impaired cortical bone repair following injury and reduced callus formation during fracture healing [137,138].

Finally, studies in dental follicle stem cells (DFSCs) demonstrated that pharmacological inhibition of β-catenin signaling suppressed their osteogenic differentiation, suggesting that this pathway is broadly conserved across various craniofacial and skeletal progenitor cells [139,140].

### 4.3. Post-Transcriptional Regulation of β-Catenin by miRNAs and Extracellular Signals Modulating β-Catenin Signaling

The results of the subsequent studies show a significant effect on miRNA expression when comparing parathyroid hormone (PTH) with dexamethasone (DEX) after a very short treatment time and a more pronounced effect after 24 h of treatment in human osteoblasts HOB cells HOB in vitro culture [141].

Several miRNAs showing differential expression were found to target genes involved in bone metabolism, e.g., miR-30c2, miR-203, and miR-205 targeting Runx2 and miR-320 targeting β-catenin mRNA expression (CTNNB1). CTNNB1 and Runx2 levels decreased after DEX treatment and increased after PTH treatment, so we have another factor that influences the upregulation of β-catenin transcription in vitro culture [142,143].

At the post-transcriptional level, miR-483-3p has been identified as a direct regulator of DKK2 in human osteoblasts. By targeting DKK2, miR-483-3p upregulates the expression of β-catenin and cyclin D1, thereby promoting osteoblast proliferation, preosteoblast-to-osteoblast differentiation, and new bone matrix formation. Conversely, overexpression of DKK2 downregulates Wnt1, β-catenin, and cyclin D1 protein levels, while simultaneously increasing the RANKL/OPG ratio and enhancing apoptosis in osteoblasts [144].

Moreover, another molecule miR-218 directly affects sFRP2 and DKK2, which are antagonists of the WNT signaling pathway, and enhances Wnt/β-catenin signaling activity. In contrast, mimicking the Wnt/β-catenin signal enhances miR-218 expression levels, while blocking the signal attenuates miR-218 expression levels. These studies demonstrate the influence of microRNAs on the modulation of the β-catenin signaling pathway and thus a central role in osteogenic differentiation of human adipose-derived stem cells (hASCs) [145,146,147].

Improving our knowledge of the characteristics of miRNAs in osteogenesis provides an important step towards their application in bone tissue engineering and translational therapies [148].

Notably, exposure to HIV-1 proteins has been associated with a reduction in both cytoplasmic and nuclear levels of β-catenin. This downregulation is thought to contribute to impaired TCF/LEF-mediated transcriptional activity, a key downstream effect of canonical Wnt/β-catenin signaling. In vitro studies using HOBs demonstrated that suppression of Dkk1 expression following exposure to the HIV-1 envelope glycoprotein gp120 led to enhanced ALP activity, increased cellular proliferation, and reduced apoptotic cell death. These findings suggest that dysregulation of Wnt/β-catenin signaling is a central mechanism contributing to HIV-associated bone loss, with Dkk1 emerging as a likely critical mediator of this degenerative process [149,150,151,152].

In another experiment, where the role of the Wnt/β-catenin pathway in the effect of implant topography on MG63 cells differentiation was investigated, β-catenin signaling activity was shown to be enhanced depending on the type of implant surface. On a smooth surface, exogenous Wnt3a stimulates β-catenin signaling and MG63 osteoblastic cell differentiation. The results clearly indicate that implant topography regulates the product of Wnt/β-catenin pathway modulators from cells, which, in turn, activates the cellular Wnt/β-catenin pathway, enhancing osteoblast differentiation [153].

Fluoride has also been shown to activate the canonical Wnt/β-catenin pathway and alter related gene expression and β-catenin protein localization in primary cultures of mouse osteoblasts (OB cells), promoting cell proliferation. In contrast, supplementation with 2 mmol/L Ca^2+^ reversed the expression levels of genes and proteins related to the canonical Wnt/β-catenin pathway [154]. In addition, fluoride also induces the expression of the Wnt-targeted Runx2 gene. Importantly, there is a positive effect of fluoride on ALP activity and the collagen type I alpha 1 chain (COL1A1). In addition, ALP, osteonectin, and Runx2 mRNA expression were shown to be abolished by DKK-1, a Wnt/β-catenin receptor blocker. These findings suggest that fluoride promotes osteoblast differentiation via Akt- and GSK-3β-dependent activation of the Wnt/β-catenin signaling pathway in primary rat osteoblasts, promoting the process of osteoblastogenesis and β-catenin-mediated osteoblast differentiation [155].

Neural epidermal growth factor-like 1 protein (NELL-1) has been shown to be a potent activator of the Wnt/β-catenin signaling pathway in both osteoblasts and osteoclast precursor cells. In bone marrow stromal cells (BMSCs), NELL-1 enhances β-catenin expression and nuclear translocation, thereby promoting osteogenic differentiation and increasing the expression of osteoprotegerin (OPG), a critical decoy receptor that inhibits osteoclastogenesis [156,157]. Recombinant NELL-1 has been demonstrated to bind directly to β1 integrin, initiating downstream activation of Wnt/β-catenin signaling. This integrin-mediated pathway contributes to enhanced osteoblast maturation and concurrent suppression of bone resorption, positioning NELL-1 as a promising therapeutic candidate for anabolic bone regeneration therapies [156,158].

Recent in vivo studies further support the therapeutic potential of NELL-1. Systemic or local delivery of recombinant NELL-1 protein, as well as gene therapy approaches using NELL-1 overexpression vectors, have resulted in increased bone mass, improved trabecular architecture, and enhanced fracture healing in osteoporotic animal models [159,160].

### 4.4. Mechanical Loading as a β-Catenin Activator

Of great interest is the study on the effect of mechanical loading, in which osteoblastic MC3T3-E1 cells were exposed to 3400 microstrains for 5 h, which increased the expression of Wnt10B, sFRP1, cyclin D1, FZD2, WISP2, and connexin 43. The final conclusions confirm that β-catenin signaling is essential for osteoblast differentiation and activation of the Wnt/β-catenin pathway increases the sensitivity of osteoblasts/osteocytes to mechanical loading [161,162].

The researchers report that mechanical stress increases ALP activity levels, promotes matrix mineralization, upregulates the expression of osteogenic factors such as ATF4, Osx, ALP, and β-catenin, and downregulates the expression of RANKL and RANK [161]. It is reported that these bone formation regulators related to the Wnt/β-catenin signaling pathway have a direct effect on the regulation of bone formation and differentiation mainly through Wnt molecules as mentioned earlier. Doxycycline has been shown to induce the expression of Wnt7b in osteoblastic cell lines. Elevated Wnt7b expression, particularly in aging bone, enhances both trabecular and endosteal bone formation and contributes to increased bone mineral density, especially during fracture healing [163,164].

### 4.5. β-Catenin Signaling in Osteosarcoma and Bone Tumorigenesis

Wnt1 expression in mature osteoblasts and osteocytes plays a critical role in modulating the function of these cells and in maintaining skeletal homeostasis [101,102]. In contrast, oxidized phospholipids (oxPLs), which bind to LRP6, trigger its internalization via clathrin-mediated endocytosis. This process reduces the responsiveness of mesenchymal stem cells (MSCs) to osteogenic stimuli by inhibiting canonical Wnt signaling and ultimately impairing osteoblast differentiation potential [37,165].

On the other hand, numerous reports indicate the role of β-catenin in cancer cells, including osteosarcoma. The canonical Wnt/β-catenin signaling pathway is a key component of normal skeletal development and disease. Alterations in this signaling pathway have been described in human osteosarcoma. In metastatic osteosarcoma, the Wnt/β-catenin pathway promotes tumor cell invasion and migration, and β-catenin acts as a biological marker with metastatic potential. The involvement of the Wnt/β-catenin pathway in osteosarcoma development and metastasis is regulated by several factors, including hormones and alkaline phosphatase [166,167].

In human osteosarcoma cell lines, major components of the Wnt/β-catenin pathway, including Wnt3a, β-catenin, and LEF1, are upregulated compared to human fetal osteoblasts. This aberrant expression of components of the canonical Wnt/β-catenin pathway suggests a key role for canonical Wnt/β-catenin signaling in the development of bone tumors [168].

A team of researchers led by Zhi-Cai Zhang concluded that reduced expression of miR-107 in human osteosarcoma cells (HOS) results in an increased expression level of Dkk-1, which results in the inhibition of the expression of proteins associated with the Wnt/β-catenin signaling pathway. As a consequence, osteoblast differentiation is inhibited and thus it is a new therapeutic target in the treatment of osteosarcoma [169,170,171].

Downregulation of miR-31-5p expression causes Axin1 to activate the Wnt/β-catenin signaling pathway, thereby inhibiting osteosarcoma cell proliferation, invasion, and tumorigenesis [172]. The Wnt/β-catenin pathway is also inhibited by IL-24, resulting in inhibition of proliferation of osteosarcoma cell lines [173].

However, research by John Aggelidakis’ team et al. presents a novel mechanism by which biglycan, an extracellular matrix proteoglycan, enhances osteosarcoma cell growth by regulating the LRP6/β-catenin/IGF-IR signaling pathway [174,175].

The canonical Wnt/β-catenin pathway serves as a central hub integrating mechanical cues, post-transcriptional regulation by miRNAs, and diverse extracellular signals to coordinate osteoblast proliferation, differentiation, and bone tissue remodeling. In vivo studies consistently corroborate the critical role of β-catenin modulation in bone health and disease, suggesting that therapeutic targeting of this pathway holds promise for enhancing bone regeneration and mitigating bone cancers.

## 5. Conclusions

Accumulating evidence indicates that β-catenin plays a key role in regulating osteoblast differentiation and proliferation, and consequently, β-catenin and components of the Wnt/β-catenin signaling pathway control bone formation and resorption. Understanding the molecular mechanisms of β-catenin interactions, both within the signaling pathway and in the nucleus, is crucial for modulating osteoblast differentiation in vitro and in vivo. Therefore, targeted mutations, as well as gene silencing via miRNA, present novel therapeutic avenues in the treatment of bone diseases. The aim of this review is to highlight the involvement of β-catenin as a key molecule in the signaling pathway required for osteoblast differentiation under physiological and pathological conditions.

β-Catenin is a multifunctional protein whose role is highly dependent on its intracellular localization. Through its association with cadherins at the plasma membrane, β-catenin contributes to cell adhesion, while its translocation to the nucleus enables the activation of gene transcription programs that direct lineage-specific differentiation. A comprehensive understanding of the molecular mechanisms underlying β-catenin interactions is therefore critical for elucidating its role in osteoblast biology. Genes such as Runx2, Cbfa1, Osx and Dlx5 are key in the process of differentiation of the osteoblast lineage and osteoblasts in vivo.

The Wnt/β-catenin signaling pathway plays a role not only in osteoblast differentiation, but also in their maintenance and proliferation. Mutations in molecules that are a component of the Wnt/β-catenin cascade lead to osteoblast apoptosis in vitro and abnormal bone development and their diseases such as osteoporosis or osteosarcoma. Within the central domain of β-catenin, comprised of armadillo repeats, two critical binding sites have been identified that mediate interactions with transcription factors in the nucleus. These sites represent potential therapeutic targets, particularly in the context of inhibiting β-catenin-driven transcriptional programs implicated in bone cancers.

Future perspectives in β-catenin-targeted research are likely to expand beyond the current focus on cancer therapies to encompass broader applications in regenerative medicine and metabolic bone disorders. Building on emerging findings that β-catenin silencing or modulation of its upstream regulators induces apoptosis and inhibits proliferation in osteosarcoma, future strategies may involve the development of more selective modulators such as small *molecules*, *peptide*
*inhibitors*, or advanced RNA-based therapeutics that can fine-tune β-catenin activity with minimal off-target effects. Additionally, personalized approaches that leverage patient-specific molecular profiles of Wnt/β-catenin dysregulation could lead to more effective and safer therapies for osteoporosis, bone fractures, and other skeletal pathologies. Integration of β-catenin modulation with tissue engineering or stem cell-based bone regeneration strategies also holds considerable promise for enhancing osteogenic potential while maintaining tissue homeostasis.

The present review has sought to provide a comprehensive overview of the molecular mechanisms by which β-catenin regulates osteoblast differentiation, with particular emphasis on findings derived from in vitro models, including both primary osteoblastic cultures and established cell lines. β-Catenin, as a central mediator of the canonical Wnt signaling pathway, orchestrates a complex network of transcriptional programs that govern lineage commitment, matrix maturation, and functional differentiation of osteoblasts. These processes are tightly regulated through context-dependent interactions with transcription factors such as Runx2, Osx, and Dlx5, and through cross-talk with other signaling pathways including BMP.

Moreover, β-catenin’s intracellular dynamics, ranging from cytoplasmic stabilization to nuclear translocation, are critical for modulating the gene expression patterns essential for skeletal development and maintenance.

Building upon this foundational knowledge, our research group is actively investigating the influence of natural compounds specifically rosavin and salidroside on β-catenin expression dynamics and functional activity in human osteoblastic HOB cells. These phytochemicals, known for their antioxidant and anti-inflammatory properties, may represent novel modulators of osteoblast function via the Wnt/β-catenin signaling axis. By characterizing the temporal and spatial expression patterns of β-catenin in response to these compounds, we aim to elucidate potential mechanisms by which they may enhance osteogenic differentiation or protect against osteoblast dysfunction under pathophysiological conditions.

We anticipate that the outcomes of our ongoing investigations will not only expand the current understanding of β-catenin’s multifaceted role in osteoblast biology but also inform the development of novel therapeutic strategies targeting skeletal disorders such as osteoporosis, delayed fracture healing, and bone-related malignancies. Ultimately, integrating molecular insights with pharmacological modulation of β-catenin may offer promising avenues for translational bone biology research and precision regenerative therapies.

## Figures and Tables

**Figure 1 biomolecules-15-01043-f001:**
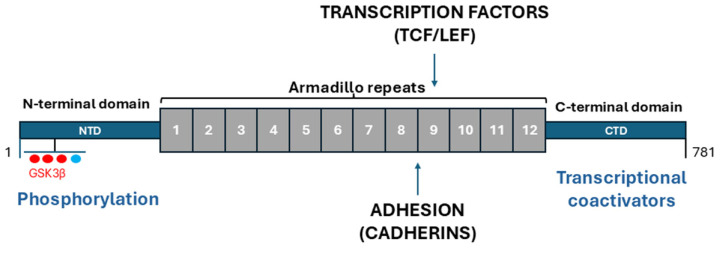
β-catenin: molecular structure and organization. The schematic image illustrates the major structural domains of β-catenin, highlighting its interaction interfaces at adherens junctions, in the cytoplasm, and within the nucleus, along with known phosphorylation sites. The human β-catenin polypeptide consists of 781 amino acid residues and is characterized by a central core of 12 armadillo repeats (ARMs) represented numerically in the diagram that serve as key platforms for protein-protein interactions. This domain mediates β-catenin binding to cadherins in the plasma membrane (stabilizing cell–cell adhesion) and transcription factors (e.g., TCF/LEF) in the nucleus, driving expression of osteogenic genes such as *Runx2*, *BMP-2*, and *ALP*. These repeats are flanked by two terminal regions, an NTD and CTD, which contribute to the regulation of β-catenin activity and binding specificity. The N-terminal domain (NTD) contains phosphorylation sites (Ser33, Ser37, Thr41, and Ser45) that regulate β-catenin stability via GSK3β-mediated ubiquitination and proteasomal degradation. The C-terminal domain (CTD) acts as a transactivation domain, interacting with transcriptional coactivators such as CBP/p300 and recruiting chromatin remodeling complexes during the transcriptional activation of Wnt target genes.

**Figure 2 biomolecules-15-01043-f002:**
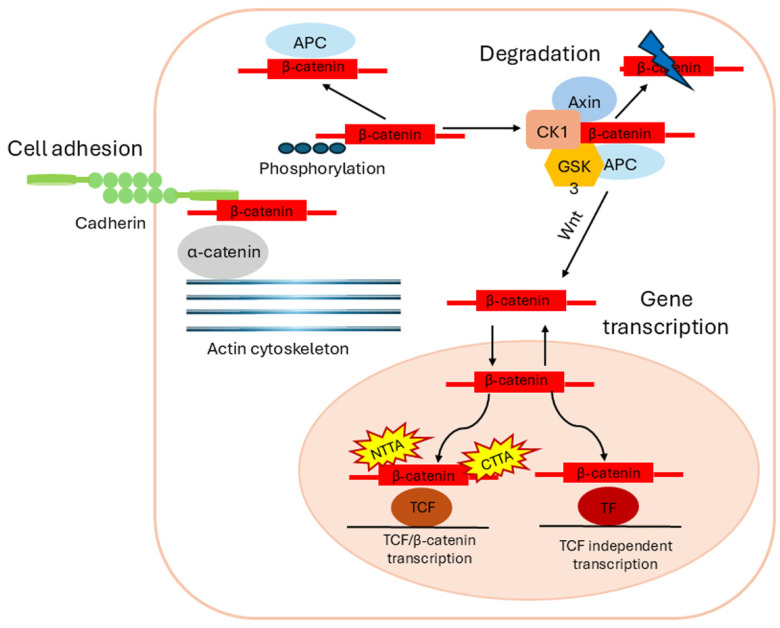
Intracellular dynamics of β-catenin. Following synthesis, β-catenin is sequestered at adherens junctions through interaction with E-cadherin, where it also associates with α-catenin, thereby indirectly influencing the organization of the actin cytoskeleton. Release of β-catenin from junctional complexes may occur through kinase-mediated signaling events or through E-cadherin downregulation. Once liberated, excess cytoplasmic β-catenin is rapidly phosphorylated by the destruction complex, targeting it for proteasomal degradation. However, a subset of β-catenin molecules may evade degradation by associating with APC in the cytoplasm, which offers transient protection. Activation of the Wnt signaling pathway inhibits the activity of the destruction complex, resulting in accumulation of cytoplasmic β-catenin. The stabilized protein is then translocated into the nucleus, where it forms complexes with TCF/LEF family transcription factors to initiate the expression of Wnt/β-catenin target genes. In addition to TCF/LEF, alternative nuclear factors can provide DNA-binding platforms for β-catenin, sometimes antagonizing canonical Wnt-mediated transcription. The transcriptional function of β-catenin within the nucleus is further modulated through regulation of its nuclear import and export. Beyond its dual role in structural cell–cell adhesion and nuclear gene regulation, β-catenin may also contribute functionally at the centrosome. Abbreviations: CTTA, C-terminal transcriptional activators; NTTA, N-terminal transcriptional activators (adapted from Valenta et al. [13]).

**Figure 3 biomolecules-15-01043-f003:**
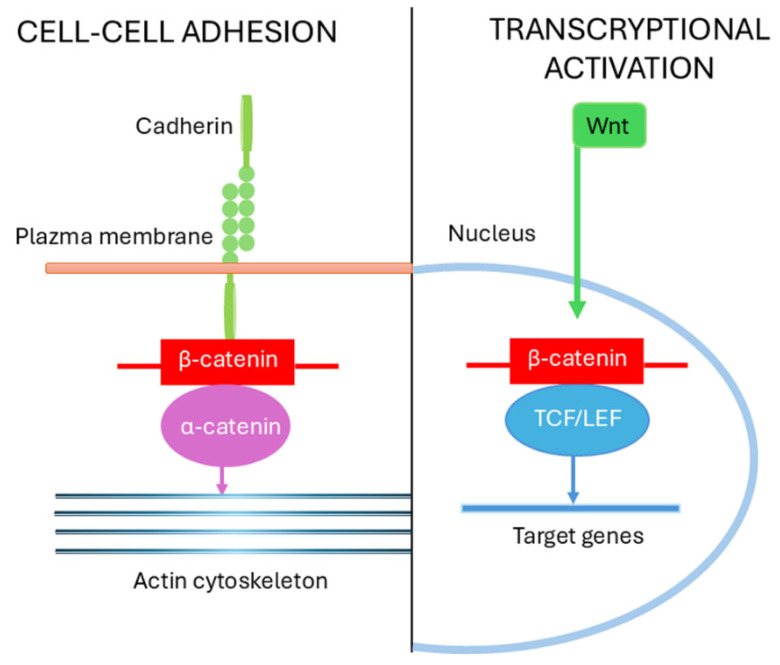
Schematic representation of the dual cytoplasmic and nuclear functions of β-catenin. On the left, β-catenin contributes to cell–cell adhesion by interacting with cadherins at adherens junctions, linking the cadherin complex to α-catenin and the actin cytoskeleton, thereby maintaining integrity and communication. On the right, upon activation of the Wnt pathway, β-catenin translocates to the nucleus where it cooperates with TCF/LEF transcription factors to regulate the expression of target genes that promote proliferation, differentiation, and extracellular matrix formation. This dual role of β-catenin underscores its importance in coordinating structural adhesion with transcriptional programs.

**Figure 4 biomolecules-15-01043-f004:**
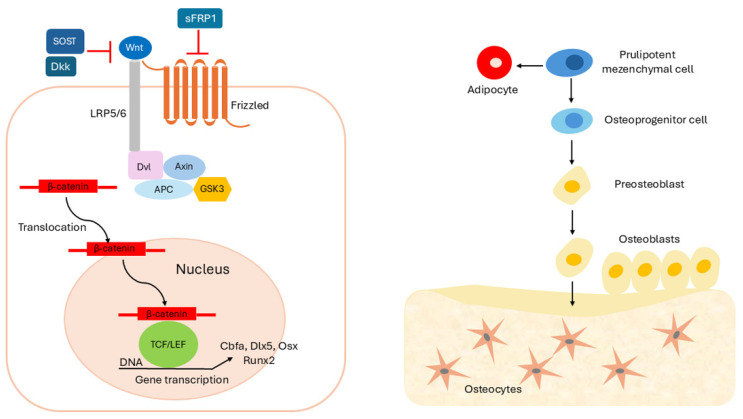
The canonical Wnt signaling pathway plays multiple roles in osteoblastogenesis, being essential for early osteoblast lineage differentiation and also involved in proliferation, maintenance, and differentiation. Interaction of Wnt with FDZ and LRP5/6 receptors induces a signaling cascade that results in the accumulation of β-catenin in the cytoplasm. β-catenin then translocates to the nucleus, where it activates transcription of target genes for osteoblastogenesis and bone formation (scheme on the **right**). Osteoblasts are derived from mesenchymal precursor cells, which also give rise to adipocytes, among others. The specification of osteoblast identity is governed by transcription factors such as Runx2 (Runt-related transcription factor 2), Cbfa1, and Osx, which are recognized as key regulators of osteogenic commitment (scheme on the **left**).

**Table 1 biomolecules-15-01043-t001:** Structural domains of human β-catenin and their primary functional roles.

Domain/Region	Amino Acid Range	Main Functions	References
N-terminal domain (NTD)	~1–140	Contains phosphorylation sites (Ser33, Ser37, Thr41, Ser45) critical for β-catenin degradation by the destruction complex (Axin, APC, GSK3β)	Valenta et al., 2012 [13]
Armadillo (ARM) repeat domain	141–664	Composed of 12 armadillo repeats forming a superhelix; mediates interactions with cadherins at the plasma membrane and transcription factors (TCF/LEF) in the nucleus	Huber & Weis, 2001 [26]
C-terminal domain (CTD)	665–781	Involved in transcriptional activation; includes a C-terminal helix and unstructured regions that interact with co-activators	García de Herreros, 2002 [24]
C-helix	~within 665–781	Stabilizes the β-catenin structure and contributes to its transcriptional activity	Gottardi & Peifer, 2008 [22]
Unstructured regions (N- and C-terminal ends)	1–~50 (N-term), ~750–781 (C-term)	Flexible regions facilitating interactions with various regulatory proteins	Xing et al., 2008 [18]

## Data Availability

Not applicable.

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
