# Peer review of "β-Catenin: A Key Molecule in Osteoblast Differentiation"

_biomolecules, 2025, doi:10.3390/biom15071043_

Round 1
Reviewer 1 Report
Comments and Suggestions for Authors
Journal: Biomolecules (ISSN 2218-273X) Manuscript ID: biomolecules-3717248 Type: Review Title: β-catenin: A Key Molecule in Osteoblast Differentiation Authors: Edyta Wróbel * , Piotr Wojdasiewicz , Agnieszka Mikulska , Dariusz Szukiewicz Section: Molecular Biology The paper provides a detailed and technically rich assessment of β-catenin's role in osteoblast biology, specifically within the Wnt signaling framework. It effectively outlines the structural domains, signaling pathways, and regulatory molecules involved in osteoblast differentiation. The manuscript may benefit from a clearer explanation of its uniqueness, a more compact format, and a greater understanding of context-specific β-catenin regulation. Redundancies in pathway explanations, inconsistencies in citation style, and small grammatical errors should all be rectified to improve readability and consistency. Overall, the article is useful and well-referenced, but it needs to be revised for clarity, structure, and uniqueness. Recommendation: Make major revisions before publication. 1: What particular research gap does this review seek to fill, and how does it offer a fresh viewpoint on the function of β-catenin in osteoblast differentiation? Need to mention in the abstract. 2: There is insufficient explanation in Figure 1. Domains with structural context are hardly mentioned in literature.Using the illustration, include an explanation of domain interactions. 3: How does β-catenin distinguish between its adhesive role at adherens junctions and its transcriptional role in the nucleus, and what post-translational modifications (e.g., phosphorylation) mediate this functional switching? 4: Overuse of technical terms without a concise explanation. restricts access for readers in the biosciences. Include a paragraph that summarizes and simplifies the mechanics of β-catenin. 5: For the dual adhesion/signaling function of β-catenin, there is no figure.The text's explanation of cadherin-linked interactions is intricate. Include a figure that distinguishes between nuclear and cytoplasmic functions. 6: The text lacks structural resolution and binding kinetics to explain partner competition, however it does identify mutual exclusivity and overlapping binding on ARM repeats (lines 127–132). To improve the mechanistic rigor of the review, it would be helpful to clarify how β-catenin prioritizes interactions under normal versus pathological conditions. 7: Add a short “Methodology of review” paragraph. 8: Several redundant Descriptions of WNT pathways. For instance, the pathway is reiterated in Section 3 and again in Section 4. Early on, combine into a single master explanation. 9: Table 1 has no justification or source. There are no inclusion criteria for molecules. Include a legend or citations to support your inclusion. 10: “β-catenin plays a main role…” Slightly awkward phrasing “β-catenin plays a central role…” 11: The review mentions several Wnt ligands in osteoblastogenesis (lines ~285–300), but it doesn't explain how, in spite of convergent on β-catenin, these ligands produce distinct transcriptional outputs. The reader's understanding of context-dependent signaling would be enhanced by investigating possible co-activator specificity or epigenetic modulators.
Reviewer 2 Report
Comments and Suggestions for Authors
The manuscript entitled “β-catenin: A Key Molecule in Osteoblast Differentiation” is a narrative review that is focused on the β-catenin role in osteogenesis. Overall, the review is well-written and presents some illustrations and tables. The main concern is related to novelty – searching in PubMed, more than 60 reviews related to osteogenesis and β-catenin were published in the last 5 years. Therefore, the authors must highlight the novelty of this study and its differential factors. Another concern is the structure of sections 3 and 4, which the authors can find below.
Abstract:
The text is well-written, but in the reviewer’s opinion, the authors should further develop it by incorporating some content from the summary (at the end of the manuscript). The abstract should include the main findings, such as the most important genes and signaling pathways. As it stands, the abstract just invites the reader to access the full text.
Introduction:
Good line of thinking and concise, passing through its role, history, osteogenic context, and objectives.
Lines 51-56: The authors included two questions to provide context for beta-catenin mechanisms. However, the reviewer believes these questions should be avoided in scientific writing, except in systematic reviews with a well-formulated question. Therefore, the reviewer suggests restructuring this paragraph.
Section 2:
The authors divided the content into molecular structure, adhesion molecule, cytoplasmic interactions, and nuclear activity. The subdivision is adequate and serves didactic purposes.
Lines 115-124: Add some references in the text.
Sections 3 and 4:
Lines 264-272: The authors describe the beta-catenin degradation and antagonists and move to the aberrant activation of the pathway. Therefore, the normal activation and the physiological pathway are not described. Additionally, there is no paragraph division. The reviewer suggests some improvements in the line of thinking of this part.
The reviewer finds the “in vitro studies” section peculiar because most of the references in sections 2 and 3 are from in vitro studies and their findings. Furthermore, the absence of an “in vivo” or “clinical studies” section hinders the proposed division.
Some of the presented content in these sections is repetitive. Thus, taking together the previous comments, in the reviewer’s opinion, the present study should be divided differently. As a suggestion, subsections can be added, focused on superfamilies (BMPs), siRNAs, classic genes (Runx2, COL, OPG), osteoporosis (RANKL), and cancer.
Section 5:
As a suggestion, rename this section as conclusions
Lines 613-616: In the reviewer’s opinion, this text should be removed, and a more generic “future perspectives” should be added.
Reviewer 3 Report
Comments and Suggestions for Authors
The manuscript provides a comprehensive overview of the role of β-catenin in osteoblast differentiation, highlighting both its structural biology and its functional implications in bone physiology and pathology. The review is thorough, referencing a wide range of recent studies and offering valuable insights into the molecular mechanisms underlying β-catenin’s dual roles in cell adhesion and signal transduction. The discussion of canonical Wnt/β-catenin signaling in osteoblastogenesis is particularly well-organized and informative, and the inclusion of both in vitro and in vivo evidence strengthens the review’s scientific rigor.
However, several aspects of the manuscript would benefit from further clarification and revision. Some sections, especially those describing molecular interactions and signaling cascades, could be streamlined for clarity. Additionally, the review would be strengthened by a more critical evaluation of conflicting findings in the field, as well as a clearer distinction between established knowledge and emerging hypotheses.
[Introduction]
The author states that β-catenin is indispensable for terminal osteoblast differentiation but not for the initial transition of osteoprogenitor cells. Could the author clarify the specific experimental evidence supporting this distinction, and discuss whether this is universally observed across different model systems?
The introduction mentions the dual role of β-catenin in cell adhesion and signaling. Could the author elaborate on how the balance between these two roles is regulated during osteoblast differentiation, and whether this balance shifts under pathological conditions?
[Results/Discussion]
The review discusses the involvement of various Wnt ligands in osteoblastogenesis but notes that the specific ligand responsible for activating the pathway remains unidentified. Could the author provide more detail on current efforts or methodologies aimed at identifying these ligands in bone tissue?
The manuscript describes the competitive binding of β-catenin to different partners (e.g., cadherins, APC, TCF/LEF). Could the author discuss how post-translational modifications of β-catenin influence its binding preferences and downstream signaling outcomes?
Several in vitro studies are cited regarding the regulation of β-catenin by microRNAs and other factors. Could the author comment on the translational relevance of these findings and whether any of these regulatory mechanisms have been validated in vivo or in clinical samples?
The review mentions the interplay between Wnt/β-catenin and BMP signaling pathways in osteoblast differentiation. Could the author clarify whether this crosstalk is context-dependent, and if so, what factors determine the dominance of one pathway over the other in specific stages of bone development or disease?
[Conclusion]
The author suggests that targeting β-catenin activity presents a promising therapeutic strategy for both bone malignancies and metabolic bone diseases. Could the author discuss potential risks or unintended consequences associated with modulating β-catenin signaling in clinical settings?
The conclusion highlights ongoing research on rosavin and salidroside in regulating β-catenin expression. Could the author provide preliminary insights or expected outcomes from these studies, and how they may contribute to the current understanding of β-catenin in osteoblast biology?
Round 2
Reviewer 1 Report
Comments and Suggestions for Authors
I have reviewed the revised version of the manuscript. The authors have thoroughly addressed all the comments and concerns raised during the previous round of review.
The revisions are satisfactory, and the manuscript has been improved both in clarity and scientific rigor. I believe it is now suitable for publication and recommend its acceptance in the present form.
Best regards
Author Response
We sincerely thank the Reviewer for the positive final assessment and for their valuable comments, which have significantly contributed to the refinement of the final version of the our manuscript.
Sincerely,
Edyta Wróbel, PhD, Biology
On behalf of the authors.
Reviewer 2 Report
Comments and Suggestions for Authors
The revised manuscript entitled “β-catenin: A Key Molecule in Osteoblast Differentiation” presents notable improvements compared to the original submission. The authors have diligently addressed all the reviewer comments and concerns, particularly the novelty factor, by incorporating a paragraph in the introduction that elucidates the differential factors. Furthermore, section 4 has undergone a significant enhancement, improving the readability of the text. As a final suggestion, the reviewer suggests either omitting the methodology section (lines 105-112) or creating a separate section for it, especially considering that this is a narrative review. Overall, the reviewer commends the authors for their work.
Author Response
We sincerely thank the Reviewer for the positive final assessment and for their valuable comments, which have significantly contributed to the refinement of the final version of the our manuscript.
In response to the Reviewer’s final suggestion, we would like to clarify that the methodological description (lines 105–112) was added following the request of another Reviewer.
In accordance with the Reviewer’s suggestion, the methodology has been separated into a distinct section to improve clarity and align with the narrative review format.
Sincerely,
Edyta Wróbel, PhD, Biology
On behalf of the authors.
Reviewer 3 Report
Comments and Suggestions for Authors
The authors have adequately addressed all the concerns raised in the previous round of review. The revised manuscript shows improved clarity in methodology, interpretation of results, and justification of the chosen models. I believe the current version meets the standards of the journal and recommend acceptance for publication.
Author Response

(The authors gave the same response as above.)
